# Mechanical and Rheological Behaviour of Composites Reinforced with Natural Fibres

**DOI:** 10.3390/polym12061402

**Published:** 2020-06-22

**Authors:** Mariana D. Stanciu, Horatiu Teodorescu Draghicescu, Florin Tamas, Ovidiu Mihai Terciu

**Affiliations:** 1Department of Mechanical Engineering, Transilvania University of Brasov, Eroilor 29, 500036 Brasov, Romania; draghicescu.teodorescu@unitbv.ro (H.T.D.); terciu.ovidiu.mihai@gmail.com (O.M.T.); 2Department of Civil Engineering, Transilvania University of Brasov, Eroilor 29, 500036 Brasov, Romania; florin.tamas@unitbv.ro

**Keywords:** natural fibre composites, mechanical properties, elastic behaviour, viscous response

## Abstract

The paper deals with the mechanical behaviour of natural fibre composites subjected to tensile test and dynamic mechanical analysis (DMA). Three types of natural fibre composites were prepared and tested: wood particle reinforced composites with six different sizes of grains (WPC); hemp mat reinforced composites (HMP) and flax reinforced composite with mixed wood particles (FWPC). The tensile test performed on universal testing machine LS100 Lloyd’s Instrument highlights the elastic properties of the samples, as longitudinal elasticity modulus; tensile rupture; strain at break; and stiffness. The large dispersion of stress–strain curves was noticed in the case of HMP and FWPC by comparison to WPC samples which present high homogeneity of elastic–plastic behaviour. The DMA test emphasized the rheological behaviour of natural fibre composites in terms of energy dissipation of a material under cyclic load. Cole–Cole plots revealed the connection between stored and loss heat energy for studied samples. The mixture of wood particles with a polyester matrix leads to relative homogeneity of composite in comparison with FWPC and HMP samples which is visible from the shape of Cole–Cole curves. The random fibres from the hemp mat structure lead to a heterogeneous nature of composite structure. The elastic and viscous responses of samples depend on the interface between fibres and matrix.

## 1. Introduction

The natural fibres, vegetal, animal or mineral, consist of sustainable resources for composite materials used both in industrial applications and building structures. The source of vegetal fibres is different parts of plants such as bust for jute, hemp, flax, ramie, kenaf, leaf for sisal, banana and manila hemp (abaca), seeds in the case of cotton, coir and oil palm, wood and grass stem [1,2,3,4]. All vegetal fibres contain cellulose, hemicellulose and lignin in various proportions. For composite materials, reinforcement can be done with continuous fibres or with short fibres, in yarn or mat form. The ratio of mechanical properties and low weight, the possibilities to design different volumetric composition with effects on the mechanical, thermal, optical and electrical properties, and the environment protection and biodegradable properties of some natural composites are demonstrated by numerous pieces of research [5,6,7].

The main criteria for assessing new natural fibre composites were usually [8,9,10]: the degree of capitalization of the vegetal raw material and of other materials; the efficiency in use of raw material sources; volume or surface density; the limit values of the resistances to different mechanical stresses (traction parallel and perpendicular to/on the surface of the plate, bending, shearing in different planes and directions, detachment, fatigue, etc.); the rigidity and elasticity of the products, expressed by the values of the longitudinal and transversal modulus of elasticity; the physical properties of lignocellulose composite materials (profile density, swelling coefficient and water absorption in different environments and periods, degree of penetration of different substances, humidity, content in crystalline substances, etc.); the ecological properties (volatile emissions, etc.); and biodegradation and recyclability. The natural fibre reinforced polymers are characterized by two stress–strain states: an elastic behaviour as a response to a fast applied force and a viscous behaviour as response to a slowly applied force. The fibre reinforcement plays an important role in elastic behaviour of polymeric composite, as the matrix contributes to the viscous behaviour. For instance, the most efficient volume fractions of fiber content is maximum 50–65 m% according to [11]. The tests to determine the flow and relaxation phenomena provide important information about the dimensional stability of the polymer [12,13]. Some studies have shown that the natural fibres used for reinforcement are thermally unstable above 200 °C, for which reason the used matrices are based on polyethylene (PE), PP, polyolefin, polyvinylchloride and polystyrene and thermosets (which can be cured below this temperature) [11,12,13,14,15,16].

Most studies on wood reinforced composites (plastic wood composites) have analysed either the influence of the particle volume fraction/resin, or the influence of the size of wood particles, or the influence of the fibre type used as reinforcing elements on the mechanical properties of composites [14,15]. Mechanical properties of wood plastic composites (WPC), determined by tensile and bending tests, show an increase of the elasticity modulus E and yield modulus at the same time with the increasing of the particle size between 0.1 mm and 0.2 mm. These mechanical parameters are lower for composites with larger particles (1.00–4.00 mm) [8,12,13,14,15,16]. The mechanical characteristics of plastic composites with wood sawdust as filler material were studied in previous research by [14,15,16,17,18]. They carried out a comparative analysis of the effects of water/seawater absorption on the degradation of mechanical properties obtained at bending, in the case of hybrid composites made of polyester resin reinforced with fiberglass E and wood flour as filler. Thus, the maximum flexural stress decreases by almost 33% after immersion in sea water of wood sawdust mixture and epoxy resin during 6572 h [15,16]. Other studies are addressed to degradation of lignocellulosic composites both by weathering, ageing or humidity, because, from a mechanical point of view, the exposure of lignocellulosic composites to different aggressive factors can produce modifications of the elastic characteristics, and can induce stress concentrations at the surface of the polymeric material (which can lead to its premature failure) [19,20,21,22,23,24]. The modifications of the elastic-dynamic properties of a composite made of small waste oak particles of various sizes and a polyester resin subjected to photo degradation by UV radiation and thermal degradations were studied by [16,25,26,27,28,29,30,31]. From the point of view of the change of the energy loss capacity (internal friction) after the exposure to UV radiations, the most stable are the specimens reinforced with 0.4 mm particles, whereas the most unstable are those reinforced with 1 mm particles [16]. Ref. [32] studied new lignocellulose composites with carbon nanotube, having improved the mechanical properties, stability and fire retardment. The mechanical behaviour of fibre-reinforced composites depends on the size of the fibres (diameter and length), their distribution in the composite structure, the strength and elasticity of the fibres, the chemical stability and thermal resistance of the matrix and the nature of the fibre-matrix interface. Depending on the main directions of stress in the final product, the layers can be oriented differently so as to ensure a multidirectional distribution of stress during applications [33,34]. Numerous studies have shown that the method of reinforcement, length and nature of the fibres influence the mechanical properties of the composite. There is a linear relationship between the increase of the fibre content and the increase of the elastic modulus of the composite [35,36,37]. The complexity of the composite behaviour consists of the differences between the rigidity of the components that lead to the development of shear stress at the interface between the matrix and the fibres [38,39,40]. The determination of the mechanical and dynamic properties of composites reinforced with natural fibres plays a key role in the different structures and applications for both the proper design of the structures and the prediction of their lifetime cycle. The finished products from natural fibre reinforced composites obtained from wood waste or hemp or flax and polyester resin are exposed to environmental and technological risks such as high relative humidity, temperature, UV radiation, vibrations, etc., reducing their resistance to different loads. The mechanical and dynamical characterisation of lignocelluloses composites based on wood particles, flax woven and polyester resin have not been investigated yet.

The aim of this paper is to examine the mechanical and dynamical properties of three different types of three categories of composites (wood particle reinforced composites, hemp mat reinforced composites and flax reinforced composite with mixed wood particles). The mechanical characterization of natural fibre composites is very important from the design and analysis as well as from the life prediction point of view, and it was obtained by a uni-axial tension test according to SR EN ISO 527-4, where the longitudinal Young’s modulus, tensile of rupture and strain at break are obtained. The polyester resin as a matrix in the studied composites is used for outdoor applications due to its high resistance to environment factors. The natural fibre composites proposed and studied in this piece of research have a good potential to be used in different parts of automotive structures as door panels, ornaments or for the indoor parts of boats. The effects of thermal degradation in terms of glass transition and heat deflection temperature were detecting by thermal dynamical mechanical analysis. Dynamic mechanical analysis (DMA) was carried out in order to provide quantitative information about the performance of material to cyclic stress and variation of temperature. Based on dynamic properties determined by DMA (stiffness, energy dissipation), the viscous-elastic properties of manufactured natural composites can be improved in order to increase the quality of composites. For the producers of composite materials and for the users of the products made from them, the present research offers information about the possibilities of use versus limitations depending on the determined mechanical properties.

## 2. Materials

Because the aim of this study is to compare the mechanical characteristics of composites reinforced with natural fibres, the sample preparation is presented for each type of specimens: wood particle reinforced composites abbreviated as WPC, hemp mat and polyurethane resin abbreviated as HMP (in longitudinal and transversal direction of mat fibres), flax and wood particle reinforced composites (abbreviated as FWPC). The specimens have the specific shape and dimensions of tensile test composite materials, according to ASRO SR EN ISO 527 and were made by a hand lay-up process.

### 2.1. Natural Fibre Composites

#### 2.1.1. Wood Particle Reinforced Composite (WPC)

The WPC specimens were obtained by mixing wood particles with polyester resin. The oak wood particles resulted from the mechanical processing of the wood logs from a Romanian factory. Oak particles were conditioned at 5–6% moisture content and their specific gravity was established by a pycnometer [41]. For the production of the lignocellulosic composites, in the first stage, the wood particles were sorted according to their size using granulometric sieves. Five classes of oak wood particles were obtained: less than 0.04 mm (coded WPC 0.04); between 0.04 ÷ 0.1 mm (coded WPC 0.10); between 0.1 ÷ 0.2 mm (WPC 0.20); between 0.2 ÷ 0.4 mm ((WPC 0.40); between 0.4 ÷ 1 mm ((WPC 1.00) and from 1 mm to 2 mm (WPC 2.00) (Table 1). From each type of grain sizes used as reinforcement, five specimens for the tensile test were prepared using the same volume fraction of 25% in a mixture with 440-M888 POLYLITE type polyester resin (Table 2), obtaining a total of 25 WPC type samples) (Figure 1a). For the DMA test, 2 specimens for the test with constant temperature and 2 specimens for the test with temperature variation (Figure 1b) were prepared from each type of wood particle reinforced composites The physical features of WPC samples for the tensile test are shown in Table 1 and for the DMA test in Table 3.

#### 2.1.2. Hemp Mat Reinforced Composites

The analysed composite material contains hemp mat and polyurethane resin (RAIGITHANE 8274/RAIGIDUR CREM), with 50% percent of reinforcing natural fibres. This type of natural fibre composite is used in the automotive industry for the interior panel of the car door. To evaluate the mechanical behaviour of composite reinforced with hemp fibres subjected to the tensile test, 5 samples on the longitudinal direction of the mat, coded HMPL were cut from the plate and 5 samples on the transversal direction of the mat, coded HMPT (Figure 2a). For the DMA test, 2 specimens for the test with constant temperature and 2 specimens for the test with temperature variation (Figure 2b) were prepared from each type of hemp mat reinforced composites. The physical features of hemp mat samples for tensile testing in Table 1 and for the DMA test in Table 3.

#### 2.1.3. Flax Reinforced Composites

For these tests, the specimens were made of polyester resin reinforced with 6 layers of flax fabric and oak wood particles with dimensions between 0.1 ÷ 0.2 mm, arranged between layers as it can be seen in Figure 3. The 6 layers of fabric have the same orientation of the warp and weft threads, respectively. The total volume percentage of reinforcement with natural fibres, in this case is approx. 30%. Samples for the tensile test were cut from the composite plate on the two main directions of the fabrics, respectively the warp direction (named FWPC_L) and the weft direction (named FWPC_T) (Figure 3a). For the DMA test, from each type of FWPC 2 specimens for the test with constant temperature and 2 specimens for the test with temperature variation were prepared (Figure 3b). The geometrical characteristics of FWPC for the tensile test are presented in Table 1 and for the DMA test in Table 3.

### 2.2. Experimental Set-Up

#### 2.2.1. Tensile Test

To determine the elastic characteristics of a material, the samples were subjected to a static tensile test. In this study, for the analysis of the mechanical behaviour of the composites, the specimens were tested on the universal testing machine LS100 Lloyd’s Instrument belonging to the Mechanical Engineering Department of Transilvania University of Brașov. The specimens were loading with a constant speed of 1 mm/min until breaking. The elongation was measured simultaneously with loading using extension device (Figure 4a).

For data acquisition, the Nexygen Plus software was used. After the tensile tests (according to SR EN ISO 527-4), the characteristic curve, the specific deformation, the longitudinal elastic modulus, the rupture tension of each reinforced composite were determined, and on the basis of the load curves, the average deformation energy for each type of sample was calculated. The fracture of samples was analysed with optical devices.

#### 2.2.2. Dynamic Mechanical Analysis

The rheological characteristics of the natural fibre composites were measured with the dynamic mechanical analysis by using the Dynamic Mechanical Analyzer DMA 242C Netzsch Germany at the Institute of Research and Development for Technical Physics in Iaşi. The method is based on ASTM D7028-07 which covers the procedure for the determination the glass transition temperature of polymer composites under the flexural oscillation mode. Thus, the complex modulus E *, with its two components (the conservation modulus E’ and the loss modulus E”) and the damping factor tan δ were determined in two cases: under isothermal conditions (T = 30 °C) for 30 min and with temperature variation between 30 to 120 °C, for 45 min. The specimens having the shape and geometry as it can be seen in Figure 4b were subjected to a flexural test. The input data were set up to 6 N for the applied force with frequency of 1 Hz. In Table 3, the specimen features for the DMA test are presented. For the DMA test, the samples have the same width and thickness as in the case of the tensile test and the length between supports is standard being set-up at 45 mm (Figure 4c).

## 3. Results and Discussion

The results in terms of qualitative and quantitative values of mechanical properties of natural fibre composites will be presented successively and then as a comparison. In the case of wood particle reinforced composites, it was noticed that, the smaller the reinforcing particles, the better the mechanical properties to traction. Nevertheless, the highest values of the elasticity modulus and of the tensile strength were obtained in the case of the composites reinforced with 0.2 mm particles. This is highlighted in the literature as well in the case of the polypropylene matrix composites (PP) [42]. The mechanical properties of the composites made of polypropylene and wood particles (WPC) in the case of tensile and bending tests show an increase in the values of the elasticity and resistance modulus simultaneous with the increase of the particle size between 0.25 and 2 mm, which is then followed by a slight decline of these values for larger particles (2–4 mm) [42]. The more elongated the particles, the more the mechanical properties increase because the contact between the reinforcing elements and the matrix occur over a larger surface. The first debonding of the matrix and the dispersed fibres occur near the breaking point. The characteristic curves for the same category of composites reinforced with wood particles did not display a large dispersion of values, the mixture between matrix and fibres being homogeneous (Figure 5).

Tensile tests performed by other researchers have shown that natural fibre fabric reinforced composite materials have major differences in the mechanical properties of traction in the direction of the warp and weft [42,43,44,45] as it can be seen in the case of HMP specimens (Figure 6a,b) and FWPC specimens (Figure 6c,d). It is known that the fibre mat is used as reinforcement to assure a quasi-isotropy of composite plates due to the random orientation of the fibres. Despite this assumption, the tests performed on hemp mat composites showed that there are significant differences between the tensile properties in the two directions of the mat (longitudinal and transversal direction). In the case of flax wood particle reinforced composites FWPC, the trend is similar to HMP regarding the mechanical properties in the weft and warp direction. Equally, it can be observed that HMP show a great spread of stress–strain curves in the longitudinal direction (Figure 6a) by comparison to FWPC which indicated a great dispersion of the curves in the transversal direction (Figure 6d).

In Figure 7, it can be noticed that the FWPC samples present a rigid behaviour compared to the HMP_L samples which behave viscously. The WPC0.20 samples are also rigid by comparison to the other wood particle reinforced composite.

The mechanical properties of tested samples in terms of the average values of the longitudinal elasticity modulus, stress at break, the strain and stiffness in percent are summarized in Table 4. In the case of WPC specimens, it can be observed that the modulus of elasticity varies from 2877 MPa, for the specimens reinforced with particles about 1 ÷ 2 mm, up to the maximum value of 4012 MPa, for the specimens reinforced with particles about 0.1 ÷ 0, 2 mm. The minimum values of the tensile strength were noted for specimens reinforced with particles about 1 ÷ 2 mm (WPC 2.0), being 19.5 MPa, and the maximum being 26 MPa, for specimens reinforced with particles from 0.1 to 0.2 mm (WPC 0.2). For the HMP specimens cut in the longitudinal direction, an average tensile strength value of 26 MPa was obtained, and in the weft direction 32 MPa, approximately 23% higher than in the longitudinal direction. In the case of the longitudinal modulus of elasticity, it is observed that, in the transverse direction, its value is approximately 63% higher than in the longitudinal direction (Table 3). For the FWPC specimens cut in the warp direction, tensile strength values between 23.9 and 27.3 MPa were obtained, and in the weft direction between 31.4 and 42 MPa. The longitudinal elasticity modulus is higher in the transversal direction (weft) by almost 95% than in the warp direction (longitudinal). Although the mechanical properties of lignocelluloses samples are relatively low, they can be used in some applications as exterior and interior products, being valuable for the possibility of integrating wood residues from processing operations or from recycled wood in the form of chips and fibres.

### 3.1. Dynamic Mechanical Analysis

#### 3.1.1. Isothermal Conditions

The rheological characteristics of the natural fibre composites in terms of complex modulus E *, with its two components (the storage modulus E’ and the loss modulus E”) and the damping tan δ were determined with the DMA. The storage modulus E’ represents the capacity of materials to withstand the applied loading, being the expression of the elastic constant of the composite. The energy dissipation due to the internal friction of the material is called loss modulus E” and it represents the viscous modulus. The ratio between the loss and storage modulus represents damping (tan δ). tan δ is an indicator of how efficiently the material loses energy to molecular rearrangements and internal friction [46]. In Figure 8, Figure 9 and Figure 10, the elastic and viscous responses of samples to dynamic loading with frequency of 1 Hz can be noticed. The capacity of WPC samples to store the deformation energy decreases slowly while increasing the time of loading. Similar behaviour is noticed in the case of the HMP samples with the mention that there are clear differences between the samples cut in the longitudinal direction compared to those cut in the transverse direction (Figure 8, Figure 9 and Figure 10b). 

Both types of samples have a similar viscous-elastic behaviour at the beginning, so that, after 10 min of stress, the samples cut in the longitudinal direction show a stiffening phenomenon compared to the samples cut in the transverse direction whose elastic behaviour decreases suddenly after 18 min of cyclic loading (Figure 8 and Figure 9c). As far as the values of complex modulus and storage modulus for each type of the tested samples are concerned, it can be noticed that the lower values are noticed in the case of the HMP sample by comparison to WPC and FWPC. Initially, there are 1120–1260 MPa and, after 5 min, they decreased by 12.5% in the case of HMP_L maintaining constant value for remaining time exposure to cyclic loading. The HMP_T indicated a higher decreasing by almost 28% after 10 min and then a stabilization of the values at around 790 MPa. In the case of the WPC samples, the complex modulus ranges between 3250 MPa (WPC0.40) and 4270 MPa (WPC0.20). It can be noticed that the values for the flexural test in a dynamic regime are similar to Young’s modulus values obtained in the tensile test. FWPC samples obtained the highest values for the dynamic modulus (5900–6060 MPa) in comparison to the other types of specimens. The loss modulus increases by increasing the time exposure to cyclic loading in the case of WPC and FWPC samples while the HMP samples indicated a decrease of this viscous component (Figure 10). This behaviour is due to the type of matrix: the WPC and FWPC, which contain as matrix polyester resin, indicated a slightly higher network density and it is slightly more cross-linked than the HMP sample based on polyurethane resin. The minimum values of energy dissipation due to internal frictions are indicated in the case of HMP samples. In the longitudinal direction (HMP_L), the viscous modulus decreases by 28% during the cyclic loading and, in the transversal direction (HMP_T), the decrease is 50%. For FWPC, the internal frictions increase by increasing the exposure time to loading; the overall value varied from minimum 465 MPa to maximum 560 MPa (in the case of FWPC_T) and from minimum 490 MPa to maximum 565 MPa (in the case of FWPC_L). For the WPC samples, the loss modulus varied in accordance with the wood particle sizes: the highest value of energy dissipation is recorded for smaller wood particles (WPC0.04). Regarding the variation of loss modulus, two groups can be noticed: WPC0.04, WPC0.10, and WPC0.20 indicated an increase in internal friction and WPC1.00, WPC2.00, WPC0.40 indicated a slight decrease of the viscous modulus.

The damping tan δ as a ratio between the loss and the storage modulus is an expression of the energy dissipation of a material under cyclic load, and it depends on the interface and adhesion between fibres and matrix. Any rigid material is characterized by a high damping value, whereas any ductile material indicates a low damping value [47]. In this sense, the damping tan δ for the HMP samples tends to decrease by increasing the loading time, since, for the WPC and FWPC samples, the damping increases (Figure 11).

#### 3.1.2. Temperature Variation

The viscous-elastic behavior of composites is emphasized by the temperature variation during cyclic loading. The stiffness and rigidity stability of composites at a certain temperature can be observed on the storage modulus curves. Thus, similar behavior between the WPC and FWPC samples can be noticed in Figure 12a,c: the glassy region presented between 30 °C and 40 °C characterized the rigidity of composites due to polymeric chains; in the second region (40–80 °C), the storage modulus decreases drastically because by increasing the temperature, the internal friction in polymeric chains is accelerated leading to a rubbery region. The HMP composite behavior differs, their stiffness being affected right from the start of the test and decreasing by temperature increase (Figure 12b). Despite this behavior, the reinforcement with hemp fibers and also the reinforcement with flax fabric lead to higher values of storage modulus by comparison to wood particle reinforced composites.

Figure 13 presents the effect of reinforcement on the damping as a function of temperature at 1 Hz frequency. From this type of chart, the glass transition temperature (*T*g) can be extracted from the peak of damping variation curves. 

According to [48,49,50], lower damping values represent the improved interfacial interaction as it can be noticed in the case of FWPC and HMP (Figure 13b,c), while a higher damping value is recorded in the case of poor interfacial adhesion as it can be observed in Figure 13a—the WPC samples. Thus, the *T*g for the WPC samples is around 78 °C (for 2.00 mm and 1.00 mm wood particle sizes) and 82 °C for smaller wood particles. The *T*g for FWPC is lower than the one for WPC, being 70 °C. The HMP samples indicated a different behaviour. In Figure 14, the Cole–Cole charts are shown in order to analyse the connection between stored and loss heat energy for the studied samples. As [49,50,51] highlights, this kind of plot is useful to interpret the modification of viscous-elastic material with different reinforcement, as it is illustrated in Figure 14a—the WPC samples. The mixture of wood particles with polyester matrix leads to relative homogeneity of composite by comparison to the FWPC and HMP samples which is visible from the shape of Cole–Cole curves. The random fibres from the hemp mat structure lead to a heterogeneous nature of the composite structure.

### 3.2. Fracture Analysing

In Figure 15, the ways of breaking of the tested composites are shown. Thus, it is noticeable that the breakage of the composite reinforced with wood particles (WPC) is produced through the simultaneous destruction of the matrix and of the dispersed fibres (Figure 15). In the case of the HMP samples, the matrix is fractured first; then, the dispersed fibres break. The plane of fracture is obtained on the area with the minimum resistance of the interface between the matrix and the fibres or in the area where the fibres are missing or do not have a good adhesion with the matrix. For the FWPC composites which contain both dispersed fibres (wood particles) and flax fabric, the fracture is produced first in the matrix mixed with wood particles and then the layers of flax fabric fail. It is appreciated that the use of a fabric structure (flax) doubles the tensile strength compared to mat reinforcement. The mechanism of failure differs between short fibres/particles reinforcement and long fibres [52,53]. The tension stress causes interface debonding. [53,54,55,56] considered that the final interface between the short fibres and the matrix is easy to debond in the loading process. At the end of the final interface, the stress transfer from the matrix to the fibres depends on the shear stress only on the axial interface (Figure 16a). In the case of long fibre composites, the interface stress (shear stress) is higher due to the adhesion of resin to fibres (Figure 16b).

## 4. Conclusions

The mechanical behaviour of composite materials from three types of natural fibres was studied. The results demonstrated that both tensile and rheological behaviour depends on the size of fibres, disposure of fibres (randomly or fabric) and the type of matrix. The mechanical properties of natural fibres composites differ within the same type of composite. For instance, WPC0.20 recorded the higher values of Young’s Modulus and tensile strength in comparison with WPC2.00 which have the lower values. It can be concluded that the best wood particles size is 0.20 mm from mechanical point of view. In case of HMP and FWPC, there is strong relation between direction of loading and weft/yarn direction. The longitudinal elasticity modulus is higher in the transversal direction (weft) by almost 80–95% than in the warp direction (longitudinal). The flax woven reinforcement in case of FWPC leads to the best mechanical properties from all types of tested composites.

Regarding DMA results, HMP and also FWPC present higher values of storage modulus by comparison to wood particle reinforced composites although the increasing temperature produces a decreases in viscous-elastic behaviour for all types of samples. The *T*g for all tested natural fibres composite range between 65 and 85 °C, the matrix having the main role in modification of polymers stiffness.

The HMP presents a good capacity to absorb the energy of deformation, partially due to polyurethane resin and to the type of dispersed fibres. In the case of a structure made of lignocelluloses composite materials, consisting of layers reinforced with natural fibres as fabrics and layers reinforced with wood particles, by adding an additional layer reinforced with wood particles on the visible surface of the panels. Thus, structures with superior aesthetic properties can be made which no longer require coating with other materials. Although the structure is no longer symmetrical, it has visible surfaces with natural textures whose colour can be changed by using wood particles of different species.

The further work will focus on simulation of mechanical behaviour of complex structures made from natural fibres composites using the elastic properties determined in experimental tests and predicting the stress and strain states of them.

## Figures and Tables

**Figure 1 polymers-12-01402-f001:**
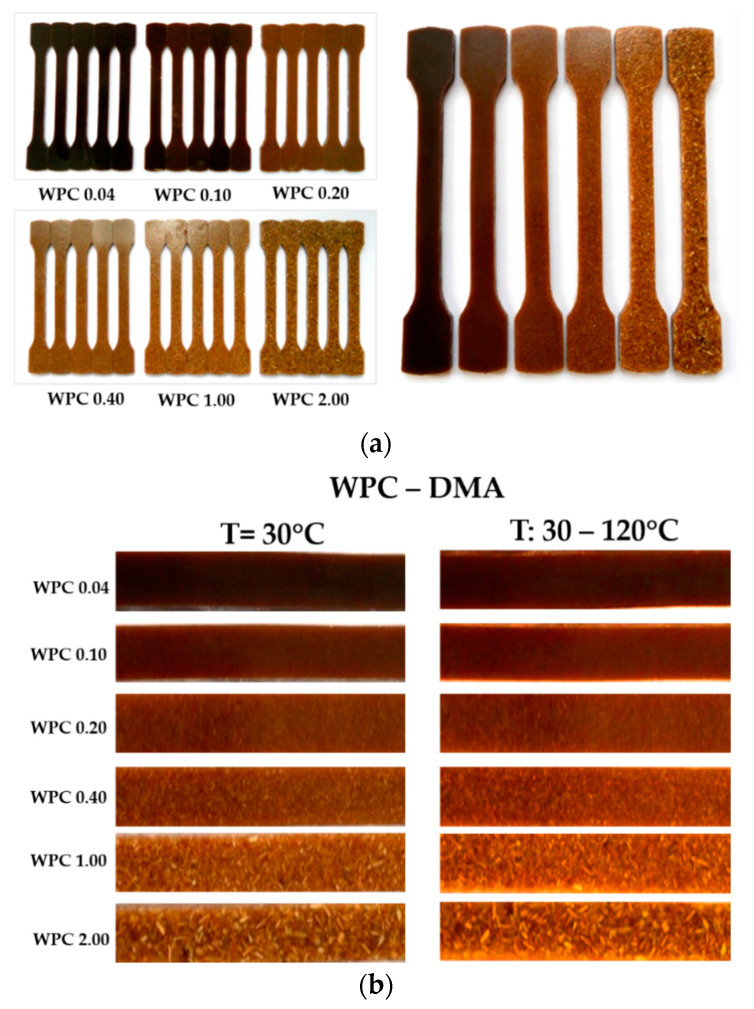
The specimens made from wood particles and polyester resin: (**a**) specimens for the tensile test; (**b**) samples for DMA.

**Figure 2 polymers-12-01402-f002:**
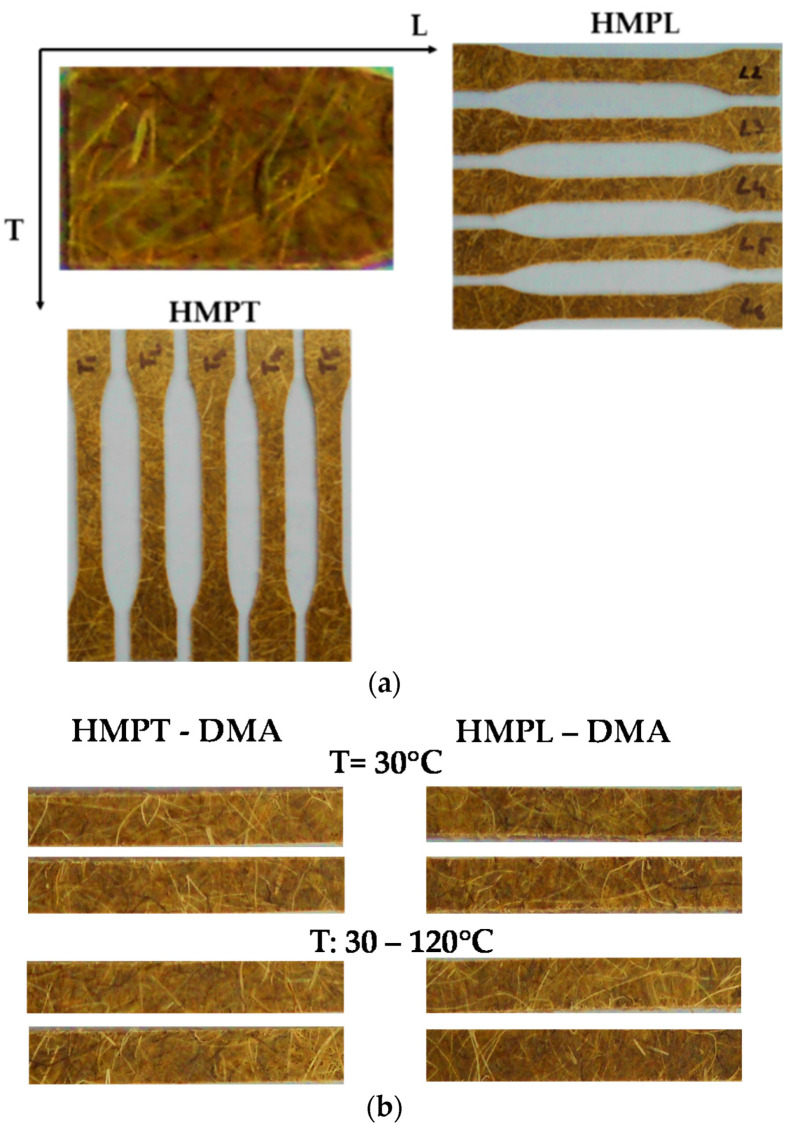
The specimens made from hemp mat reinforced and polyurethane resin: (**a**) geometry for tensile test: HMPL—sample cut in longitudinal direction; HMPT—sample cut in transversal direction; (**b**) geometry for the DMA test: HMPL—sample cut in longitudinal direction; HMPT—sample cut in transversal direction.

**Figure 3 polymers-12-01402-f003:**
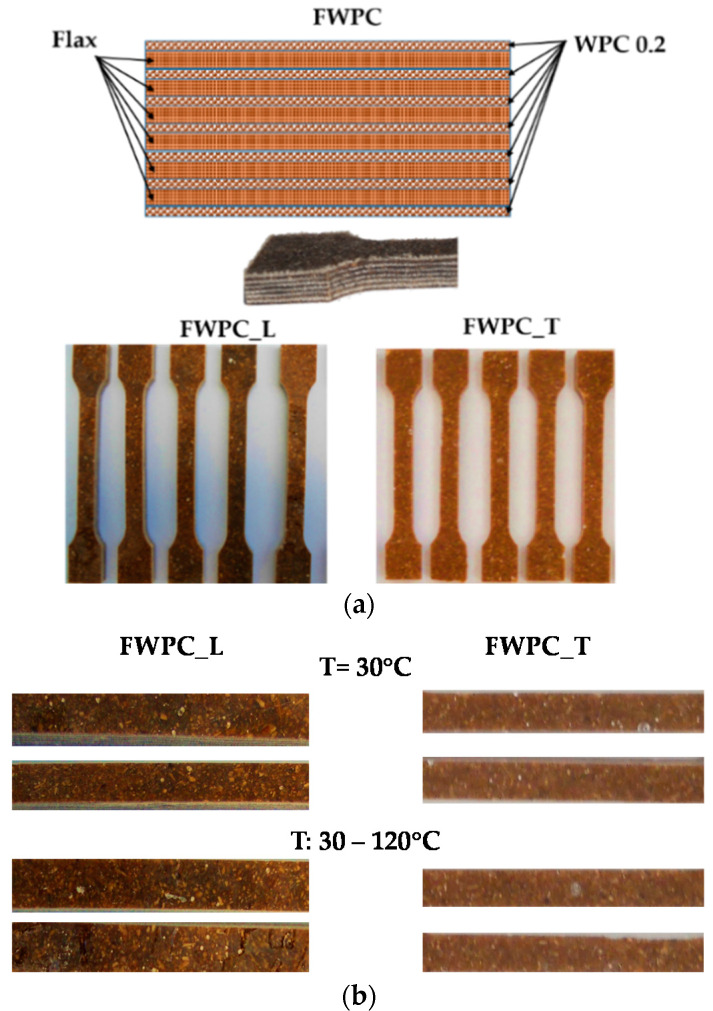
The specimens made from flax and wood particles reinforced and polyester resin: (**a**) samples for tensile test: FWPC_L—samples cut in longitudinal direction; FWPC_T—samples cut in transversal direction; (**b**) samples for DMA test.

**Figure 4 polymers-12-01402-f004:**
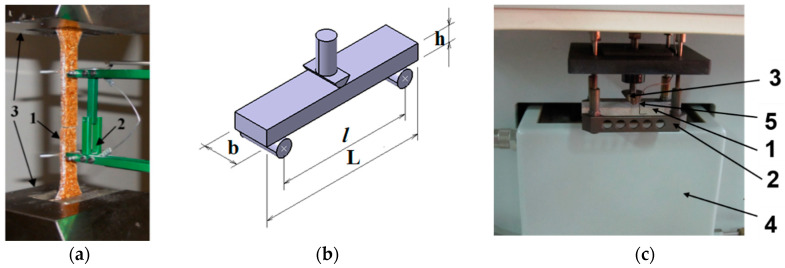
The experimental devices: (**a**) sample WPC 2.00 during the tensile test (Legend: 1—sample; 2—extension device for the elongation measurement; 3—tensile test machine jaws); (**b**) experimental set-up for the flexural test (3 points bending); (**c**) the DMA equipment (Legend: 1—sample; 2—sample supports; 3—loading device; 4—conditioner chamber; 5—temperature sensor).

**Figure 5 polymers-12-01402-f005:**
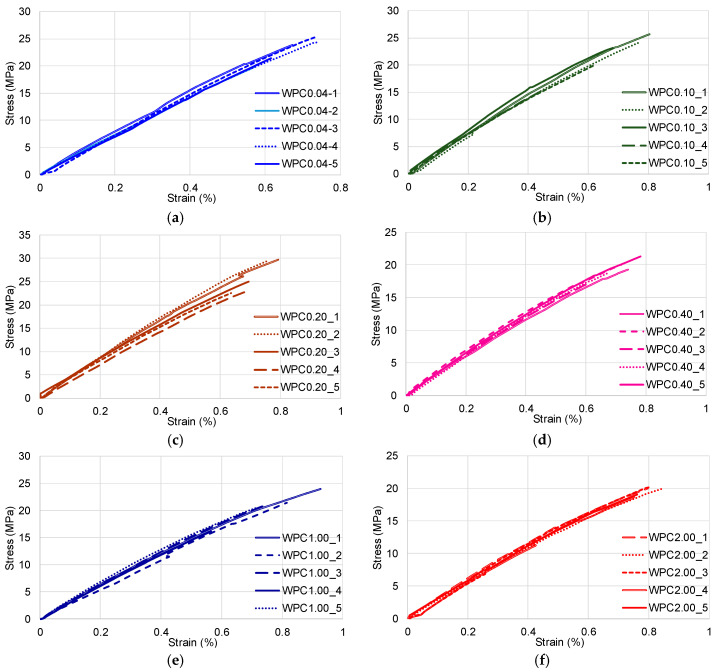
Characteristic curve stress–strain of wood particle reinforced composites: (**a**) stress–strain characteristic curves of WPC0.04 specimens; (**b**) stress–strain characteristic curves of WPC0.10 specimens; (**c**) stress–strain characteristic curves of WPC0.20 specimens; (**d**) stress–strain characteristic curves of WPC0.40 specimens; (**e**) stress–strain characteristic curves of WPC1.00 specimens; (**f**) stress–strain characteristic curves of WPC2.00 specimens.

**Figure 6 polymers-12-01402-f006:**
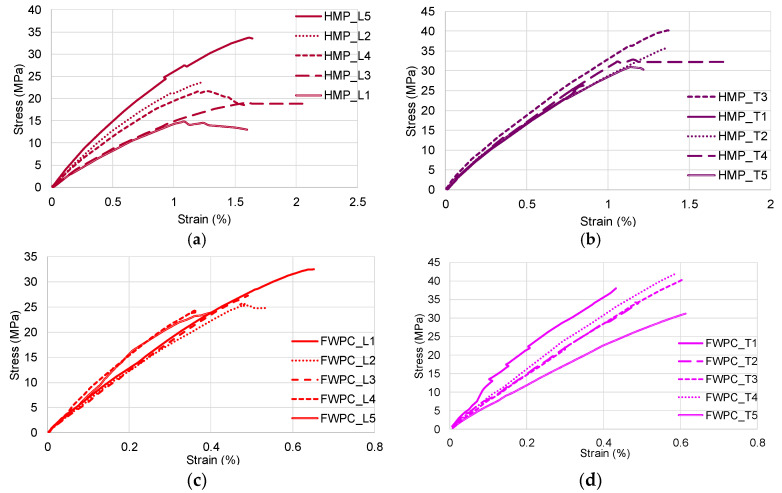
Characteristic curves stress–strain of natural fibres reinforced composites: (**a**) stress–strain curves for HMP_L samples; (**b**) stress–strain curves for HMP_T samples; (**c**) stress–strain curves for FWPC_L samples); (**d**) stress–strain curves for FWPC_T samples.

**Figure 7 polymers-12-01402-f007:**
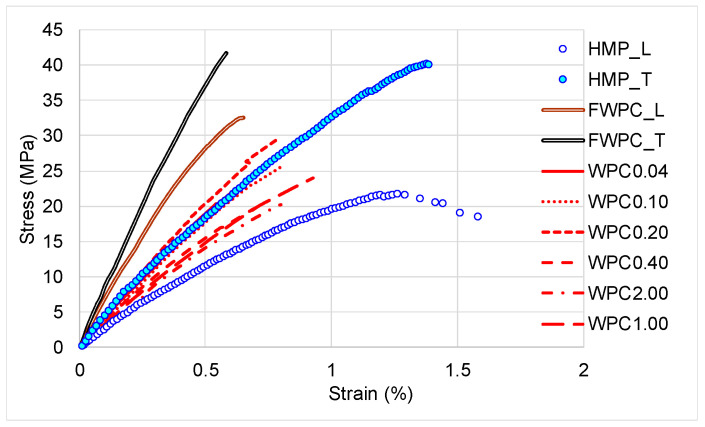
Comparison of stress–strain curves for tested samples.

**Figure 8 polymers-12-01402-f008:**
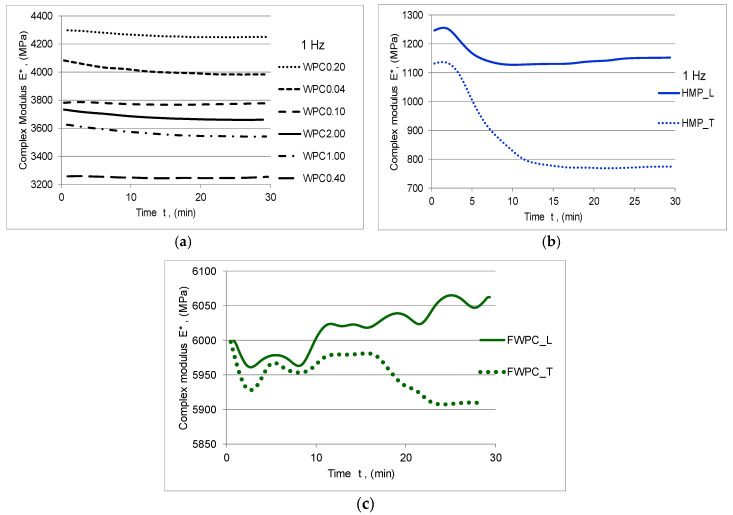
Variation of the complex modulus in time: (**a**) WPC samples; (**b**) HMP samples; (**c**) FWCP samples.

**Figure 9 polymers-12-01402-f009:**
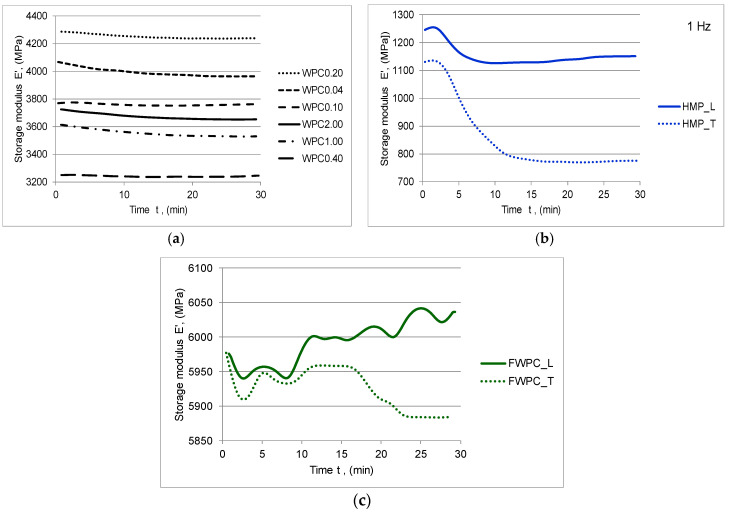
Variation of the storage modulus E’ in time: (**a**) WPC samples; (**b**) HMP samples; (**c**) FWCP samples.

**Figure 10 polymers-12-01402-f010:**
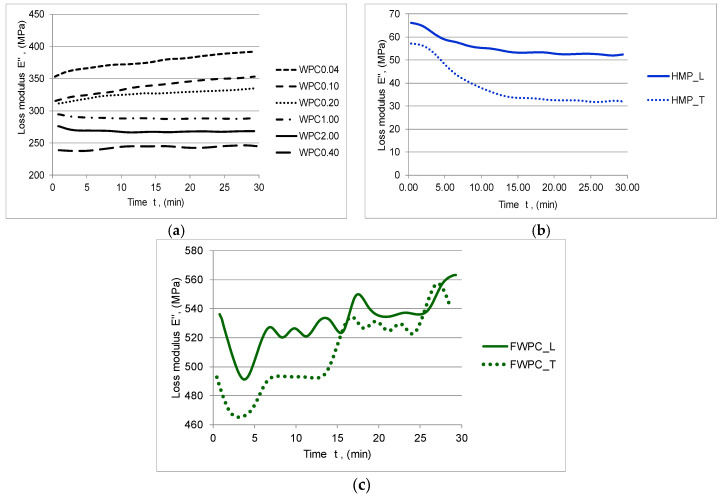
Variation of the loss modulus E’’ in time: (**a**) WPC samples; (**b**) HMP samples; (**c**) FWCP samples.

**Figure 11 polymers-12-01402-f011:**
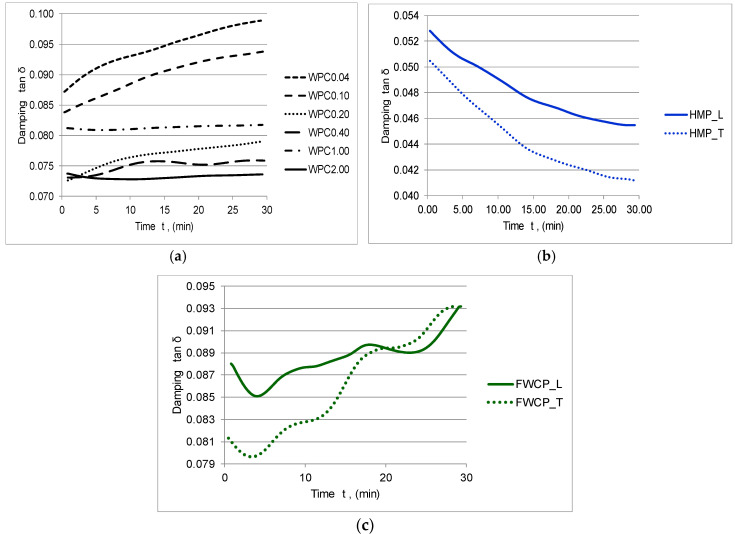
Variation of the damping tan δ in time: (**a**) WPC samples; (**b**) HMP samples; (**c**) FWCP samples.

**Figure 12 polymers-12-01402-f012:**
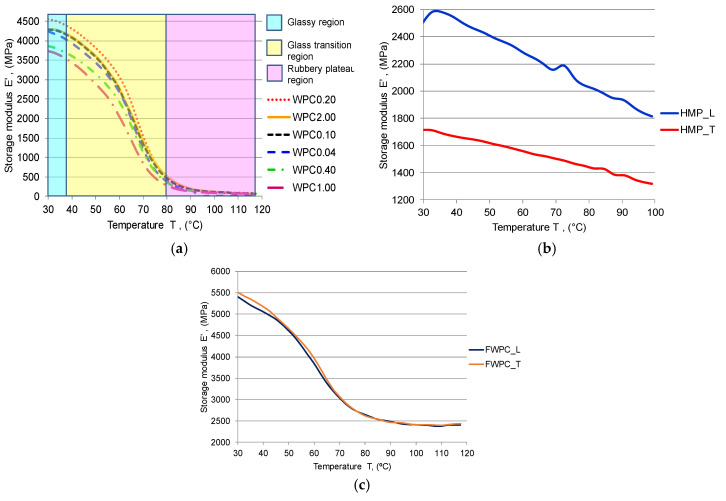
Variation of the storage modulus E’ with temperature: (**a**) WPC samples; (**b**) HMP samples; (**c**) FWCP samples.

**Figure 13 polymers-12-01402-f013:**
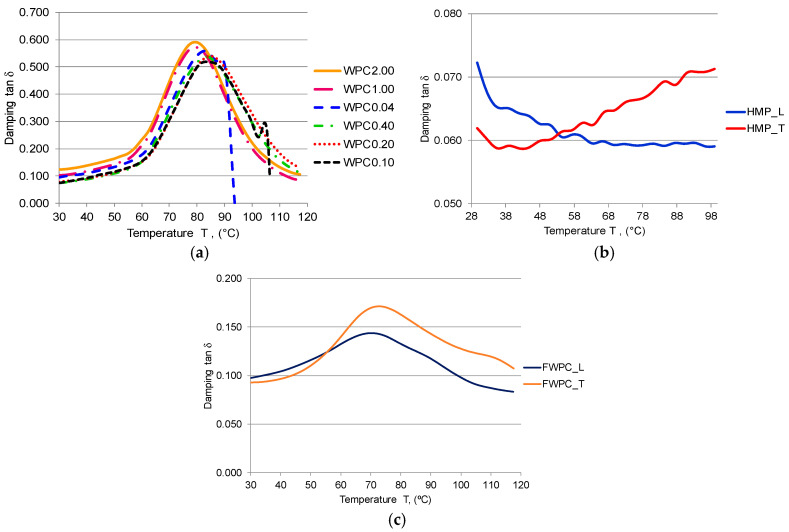
Variation of the damping tan δ with temperature: (**a**) WPC samples; (**b**) HMP samples; (**c**) FWCP samples.

**Figure 14 polymers-12-01402-f014:**
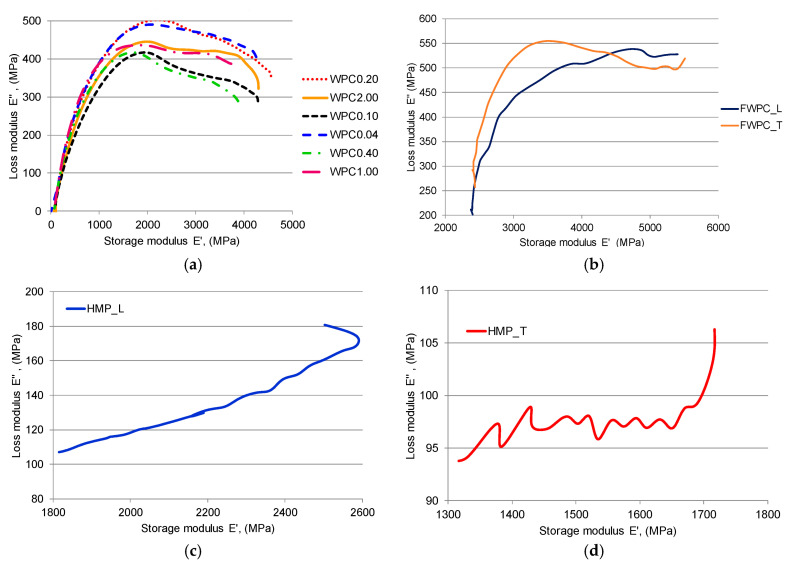
Cole–Cole plot: (**a**) WPC samples; (**b**) FWCP samples; (**c**) HMP_L sample; (**d**) HMP_T sample.

**Figure 15 polymers-12-01402-f015:**
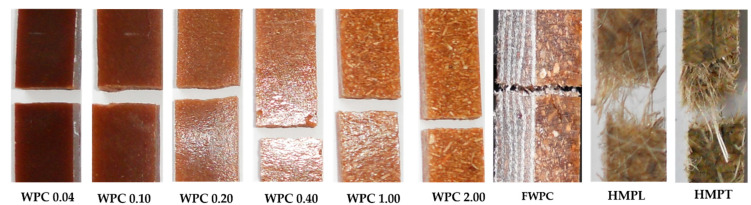
Modes of fractures recorded for all natural tested fibre reinforced composites.

**Figure 16 polymers-12-01402-f016:**
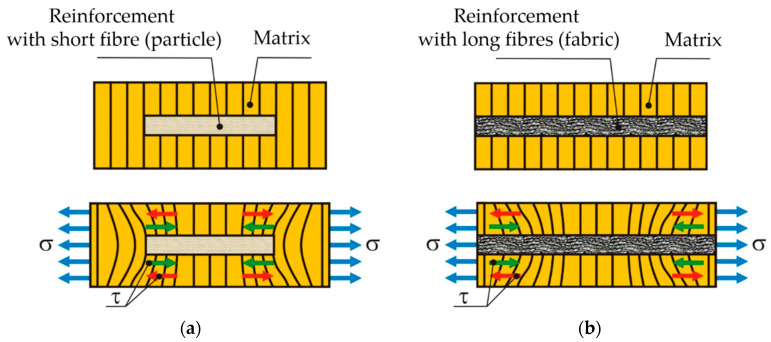
Mechanism of failure modes: (**a**) the case of short fibres reinforced composites; (**b**) the case of long fibres reinforced composites.

**Table 1 polymers-12-01402-t001:** The physical characteristics of samples for the tensile test.

Samples	No. of Samples	No. of Layers	Thickness [mm]	Width [mm]	Area [mm^2^]	Gauge Length [mm]
WPC 0.04	5	1	4.10 ± 0.10	10 ± 0.5	42 ± 1.5	50
WPC 0.10	5	1	4.10 ± 0.10	10 ± 0.5	42 ± 1.5	50
WPC 0.20	5	1	4.25 ± 0.25	10 ± 0.5	43 ± 1.5	50
WPC 0.40	5	1	4.20 ± 0.25	10 ± 0.5	42 ± 1.5	50
WPC 1.00	5	1	4.20 ± 0.20	10 ± 0.5	42 ± 1.5	50
WPC 2.00	5	1	4.30 ± 0.20	10 ± 0.5	43 ± 1.5	50
HMPL	5	1	1.80 ± 0.20	10 ± 0.5	18 ± 1.5	50
HMPT	5	1	1.80 ± 0.20	10 ± 0.5	18 ± 1.5	50
FWPC_L	5	6	6.80 ± 0.50	10 ± 0.3	69 ± 4.8	50
FWPC_T	5	6	6.60 ± 0.60	10 ± 0.4	68 ± 5.5	50

**Table 2 polymers-12-01402-t002:** The characteristics of the polyester resin type 440-M888 Polylite, at 23 °C.

Properties	Units	Value	Tested Method
Brookfield Viscosity LVF	mPa·s(cP)	1100–1300	ASTM D 2196-86
Density	g/cm^3^	1, 10	ISO 2811-2001
PH (max.)	mgKOH/g	24	ISO 2114-1996
Styrene content	% of weight	43 ± 2	B070
Curing time: 1% NORPOL PEROXIDE 1	Minutes	35–45	G020
Tensile Strength	MPa	50	ISO 527-1993
Longitudinal Elasticity Modulus	MPa	4600	ISO 5271993
Elongation	%	1.6	ISO 527-1993
Bending strength	MPa	90	ISO 178-2001
Elasticity modulus at bending	MPa	4000	ISO 178-2001
The shock resistance P4J	mJ/mm^2^	5.0–6.0	ISO 179-2001
Volume contraction	%	5.5–6.5	ISO 3521-1976
Glass transition temperature	°C	62	ISO 75-1993

**Table 3 polymers-12-01402-t003:** The physical characteristics of the samples for the DMA test (Legend: DMA operated under constant temperature T = const.; DMA operated under temperature variation T).

Samples	No. of Samples T = const./T var.	Thickness [mm]	Width [mm]	Gauge Length [mm]
WPC 0.04	2/2	4.10 ± 0.10	10 ± 0.2	40
WPC 0.10	2/2	4.10 ± 0.10	10 ± 0.5	40
WPC 0.20	2/2	4.25 ± 0.25	10 ± 0.5	40
WPC 0.40	2/2	4.20 ± 0.25	10 ± 0.5	40
WPC 1.00	2/2	4.20 ± 0.20	10 ± 0.5	40
WPC 2.00	2/2	4.30 ± 0.20	10 ± 0.5	40
HMPL	2/2	1.80 ± 0.20	10 ± 0.5	40
HMPT	2/2	1.80 ± 0.20	10 ± 0.5	40
FWPC_L	2/2	6.80 ± 0.50	10 ± 0.3	40
FWPC_T	2/2	6.60 ± 0.60	10 ± 0.4	40

**Table 4 polymers-12-01402-t004:** Average values of elastic characteristics obtained after the tensile test. Legend: E—longitudinal elasticity modulus; STDV—standard deviation; σ_r_—tensile of rupture; ε_r_—percentage strain at break; k—stiffness.

Samples	E (MPa)	STDV E (MPa)	σ_r_ (MPa)	STDVσ_r_ (MPa)	ε_r_ (%)	k (10^6^ N/mm)
WPC 0.04	3626	218	23	2	0.920	0.003052
WPC 0.10	3693	181	22	4	1.261	0.003124
WPC 0.20	4012	328	26	4	0.011	0.003518
WPC 0.40	3109	138	19	2	0.014	0.002762
WPC 1.00	3041	260	21	3	0.011	0.002683
WPC 2.00	2877	85	20	1	0.010	0.002589
HMPL	3086	934	26	9	1.550	1.110
HMPT	5005	569	32	5	1.199	1.802
FWPC_L	8586	1247	27	3	0.377	7.397
FWPC_T	16700	3500	38	3	0.37	8.854

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
