# Peer review of "Mechanical and Rheological Behaviour of Composites Reinforced with Natural Fibres"

_polymers, 2020, doi:10.3390/polym12061402_

Round 1

Reviewer 1 Report

The research presents an analysis of the mechanical and rheological behaviour of natural fibres reinforced composites.  Three different fibres were testes, Wood particles with different sizes, hemp map, and flax reinforced composite mixed with wood particles. The last one was found to be the one with a higher elastic modulus and stiffness. The article need improvement in the grammar. 

The aim of the paper is stated as " examine mechanical properties mechanical and dynamical of three different types of reinforced composites". This aim must be enhanced, by mentioning why and what will be doing, and why the methods used for the test has been chosen. 

comments: 

1. the text has grammar issues, please do a proof reading of the document. 

2. Page 5 ,line 139: factordelta, there must be a space between delta and tan.

3. For the tensile test, the standard SR EN ISO 527-4 was used, which standard was  used for the dynamic analysis?

4. What are the dimensions of the specimens for the dynamic analysis, specimens for tension and those for bending tests have different configuration, but there is no information regarding this. 

5. In he dynamic analysis, What were the force applied and the frequency? what was the time used for the energy dissipation tests?

6. Page 8, must be tan delta

7. please enhance the conclusions based on the evidence obtained from the tests, Although the aesthetics of the material might important, is not relevant and was not the case of study on this document. The research is focused on the mechanical properties, please expand on this.

Author Response

First we would like to thank the reviewers for carefully going through the manuscript and providing helpful suggestions for its improvement. Thanks to their constructive comments, we are able to present clearly and better version than the original manuscript. All the comments of the reviewers have been considered. In particular, the following changes have been made according to the reviewers' suggestions, highlighted by yellow color in the manuscript.

Reviewer 2 Report

  • In Introduction, author described past work, but little comment on the contribution and shortcoming. Author need to provide critical comments.
  • Are we in a regime which could apply to a real physical system?
  • And how is this appropriate assumption in real applications?
  • The range of defined parameters needs to be added.
  • Please highlight how the work advances or increments the field from the present state of knowledge and provide a clear justification for your work.
  • In the conclusion, please show how the work advances the field from the present state of knowledge. Please provide a clear justification for your work in this section, and indicate uses and extensions if appropriate. Moreover, you can suggest future experiments/simulations and point out those that are underway.
  • The text needs to be checked and revised by a native speaker or a language expert. You may consider (at your own cost) the use of a possible professional copyediting service
  • The conclusion section has to be rewritten doing an effort to remark the main findings rather than summarizing the article content.

Author Response

(The authors gave the same response as above.)

Round 2

Reviewer 2 Report

accepted